# Validity and Reliability of a New Test of Change of Direction in Fencing Athletes

**DOI:** 10.3390/ijerph17124545

**Published:** 2020-06-24

**Authors:** Hichem Chtara, Yassine Negra, Helmi Chaabene, Moktar Chtara, John Cronin, Anis Chaouachi

**Affiliations:** 1Tunisian Research Laboratory “Sport Performance Optimisation”, National Center of Medicine and Science in Sports, Tunis 1004, Tunisia; chtarahichem@gmail.com (H.C.); chaouachi_anis@hotmail.com (A.C.); 2Research Unit (UR17JS01) “Sport Performance, Health & Society”, Higher Institute of Sport and Physical Education of Ksar Saîd, University of “La Manouba”, Tunis 2010, Tunisia; yassinenegra@hotmail.fr; 3Division of Training and Movement Sciences, Research Focus Cognition Sciences, University of Potsdam, Am Neuen Palais 10, 14469 Potsdam, Germany; chaabanehelmi@hotmail.fr; 4High Institute of Sports and Physical Education, Kef, University of Jendouba, Tunis 8100, Tunisia; 5Sports Performance Research Institute New Zealand, AUT University, Auckland 1010, New Zealand; john.cronin@aut.ac.nz; 6PVF Football Academy, Hang Yen, Văn Giang 160000, Vietnam

**Keywords:** criterion validity, sport-specific testing, change of direction, combat sports

## Abstract

The aim of this study was to validate a new test of change of direction (COD) for fencer athletes and to establish its relationship with selected measures of physical fitness. Thirty-nine fencer athletes participated to this study (age: 20.8 ± 3.0 years). They performed the new specific fencing COD test (SFCODT) on two separate occasions to establish its reliability. In addition, assessment of COD, jumping ability (i.e., squat jump, countermovement jump, five jump test), sprint time (e.g., 5-m, 10-m and 20-m), isokinetic concentric and eccentric quadriceps, and hamstring force tests were assessed. To establish SFCODT’s construct validity, two subgroups were identified based on their international and national fencing results: High- vs. low-ranked fencer athletes. Reliability, validity, and sensitivity of the SFCODT were established from the intraclass correlation coefficient (ICC), typical error of measurement (TEM), smallest worthwhile change (SWC), and receiving operator characteristic (ROC) analysis. The ICC of SFCODT was excellent at >0.95, and the TEM was < 5%. Based on the usefulness analysis, the ability to detect small performance changes can be rated as *“good”* in fencer athletes (SWC > TEM). SFCODT was very largely associated with the COD test and moderate to very large associated with jumping ability, sprint time, and isokinetic strength. High-ranked fencer athletes were better than low-ranked fencer athletes on SFCODT (*p* < 0.01). The area under the ROC curve was 0.76. In conclusion, the SFCODT is a highly reliable, valid, and sensitive test. Therefore, the SFCODT could be used by practitioners to evaluate specific CODS performance in fencer athletes.

## 1. Introduction 

Fencing is an open-skilled combat sport between two athletes who fence each other using one of three types of weapons (i.e., foil, sabre, and epee), each contested with different rules. Fencing activity is intermittent and involves a series of high-intensity actions (e.g., attack), with changes of direction (COD) (mainly back and forth displacements and lunges) interspersed by low-intensity movements with various recovery durations [1]. It is worth noting that the lunge represents the most common form of attack in fencing [1,2,3]. These lunges are commonly delivered after numerous feints and/or changes of direction typically used to escape an opponents’ hit [3,4]. Roi and Bianchedi [4] showed that fencer athletes cover between 250 m and 1000 m, attack 140 times, and change direction nearly 400 times in women’s epee and around 170 times in men’s epee and foil. In sabre, fencer athletes perform, on average, 21 lunges, 7 changes of direction, and 14 attacks per bout [2]. In view of the large number of COD occurring, fencing could be described as combat sport in which an athlete’s agility level is a strong determinant of success [1,5,6]. High and low technical ability levels of female athletes was compared, where Roi and Pittaluga [7] claimed a significantly large number of COD in the group’s high ability (133 ± 62 vs. 85 ± 25, respectively). Moreover, lunging and changing direction, which are essential for an optimal performance, are claimed to be the most frequently used actions in fencing [4,5]. In fact, previous researchers [5,8] have found that elite-level fencer athletes are faster than non-expert ones in both lunging and changing directions. 

Limited studies have addressed the assessment of fencer athletes’ change of direction (COD) through sport-specific protocols. For instance, Tsokalis and Vagenas [5] assessed COD via a shuttle test in which the fencer athletes were asked to move as fast as possible using fencing steps both in the forward and backward directions between two parallel lines (5-m in-between), covering a total distance of 30 m. The same authors reported significant relationships (*r* = −0.44 to −0.70) between their proposed test and fencer athletes’ height, countermovement jump height, and reactive strength index derived from a 40-cm drop-jump test. Despite the ability of the shuttle test to discriminate between fencer athletes of various competitive levels, the COD distance measured is clearly longer than any distance performed in a single bout [3]. Furthermore, considering that previous investigations reported an average work time of ~15 s (much of which is submaximal), 5 s [4], and 2.5 s [2] (for fencer athletes of epee, foil, and sabre, respectively), a shorter COD test seems to be required. Turner, Bishop, Cree, Edwards, Chavda, Read, and Kirby [6] attempted to overcome the aforementioned limitations and proposed a test including COD over shorter distances (4/2/2/4-m shuttle) in conjunction with a shorter whole distance (i.e., 12 m). However, the aforementioned test did not highly respond to the nature of competitions, where most actions ended up with high-intensity forward attacks [9]. In addition, in this test, fencer athletes covered the same distance forward and backward (6 m each), while it has been shown that the forward displacement was greater than the backward in competition [9]. Moreover, the test total time was ~5 s, while a previous performance analysis study indicated that the total active time was ~7.25 s [9]. 

Given the aforementioned limitations, it would seem that a new COD test that more closely mimics the demands of the sport is required. Accordingly, the aims of this study were twofold: (1) To establish the validity of a new COD test for fencer athletes, and (2) to investigate its relationship in terms of selected measures of physical fitness (i.e., jumping, sprint time, and strength). It was hypothesized that our Specific Fencing Change of Direction Test (SFCODT) would demonstrate high levels of reliability, validity, and sensitivity in male and female fencer athletes.

## 2. Materials and Methods 

### 2.1. Experimental Approach to the Problem

Agility is an essential physical fitness ability in fencing games where rapid movement initiation and whole-body COD are determinant for successful sport participation [3,10]. The ecological validity of SFCODT was based on specific literature [3,4,9]. The accurate reproduction of the basic movement pattern of fencing game (moving with correct fencing steps forward and backward as fast as possible) has been proposed. In addition, the T-test has frequently been used as the most common protocol for testing and validate new tests of planned agility in many sports (i.e., soccer, wrestling, basketball, handball, taekwondo, skating, hockey, etc.) [11,12,13,14,15] due to its feasibility, validity, and reliability [16,17]. The T-test proved to explain COD across the sport discipline, providing empirical evidence of COD construct. For those reasons, the T-test was selected as representative of a general construct of COD and consequently stressing "construct convergent" validity in this study. Additionally, the relationship of SFCODT with sprint time (5-m, 10-m and 20-m sprint), jumping ability (squat jump (SJ), countermovement jump (CMJ), and five-jump test (5-JT)), and strength (isokinetic concentric and eccentric hamstring tests) was examined. 

### 2.2. Subjects

A minimum sample size of 38 was determined from an *a priori* statistical power analysis using G*Power (Version 3.1, University of Dusseldorf, Germany) (Faul et al., 2013). The power analysis for a Pearson correlation was computed with an assumed power at 0.80 at an alpha level of 0.05 and a large effect size (*p* = 0.5). Thirty-nine fencer athletes (21 men and 18 women) from the Tunisian national team participated in this investigation. They all had at least seven years of experience at their respective specialty. They all participated in national and international competition events regularly. The sample contained 13 athletes in each specialty. The age and the anthropometric measures of participants are shown in Table 1. When the study was conducted, these athletes represented the best fencers of their category and weapon in Tunisia based on national and/or international championships achievements and performance, and most of them were medal winners at prior previous competition events. Fencer athlete performances ranged from world-class to competitive national standard. The best fencer athletes were medalists from the FIE World Championships (02 Bronze), Olympic Games (01 Bronze), and African Championships (16 Or), European Championships (03 Or). Based on their performance, 16 fencer athletes were classified as high ranked (eight males and eight females) because they competed at the Olympic Games and the World Championships and had international FIE rankings (three Top 16, two Top 32, four Top 64, and seven Top 128). The remaining 23 were classified as low ranked (i.e., fencers who have a classification higher than 128 according to FIR ranking; 13 males and 10 females) because they participated in National and Zone Championships and had low rankings. None of the participants reported any current neuromuscular diseases or musculoskeletal injuries. The experience was conducted during the in-season period of the competitive year. After receiving a detailed explanation of the potential benefits and risks associated with the participation in the study, each participant/guardian (athletes aged <18 years) signed an informed consent form before taking part in the study. The study was conducted in accordance with the latest declaration of Helsinki (WMA 2000, Bošnjak 2001, Tyebkhan 2003), and approval was obtained from the local ethics committee of the National Centre of Medicine and Science in Sports, Tunisia.

### 2.3. Procedures

This study was conducted during the second half of the competitive season (fourth month of the season). Two weeks before the commencement of the study, all athletes attended four orientation sessions. The first session was dedicated to anthropometric measurements (age, height, body mass, body fat %). The three other sessions were used for familiarization of all tests. Height and body mass were measured using a stadiometer (Holtain Ltd., Crymych, United Kingdom) to the nearest 0.5 cm and a standard electronic scale (accurate to 0.01 kg), respectively. Skinfold thickness was measured to the nearest 0.2 mm at four predetermined sites (biceps, triceps, subscapular, and suprailiac) using Harpenden skinfold calipers (Lange, Cambridge, MA, USA). Percentage of body fat was estimated using the equations described by Durnin and Womersley [18]. All tests were completed within a two-week period and each test was separated by at least 48 hours (three separate phases). For reliability purposes during the first phase, each participant performed the SFCODT twice on two separate days. During the second phase, the criterion validity of SFCODT was examined by assessing the correlation with a generic planned agility test (T-test). A minimum of 3 min of rest was provided between trials and 30 min of rest between tests. During the last phase, the relationship of SFCODT with sprint time (5-m, 10-m and 20-m sprint), jumping ability (SJ, CMJ, and 5-JT) and isokinetic strength was examined. All tests were performed during three successive days in the following order: a) 20-m sprint time and SFCODT; b) 5-JT, SJ, and CMJ; and c) isokinetic strength. Change of direction speed, jumping ability, and sprint time tests were performed in triplicate, and a minimum of 3 min of rest were allocated between trials. The best trial of each test was used for statistical analysis. Approximately 30 min of rest was provided between tests to avoid any risk fatigue [11]. Before each session, fencers completed a 15-min standardized general warm-up including low intensity jogging, a series of dynamic stretching exercises, and multidirectional sprint in forward, lateral, and backward directions. All procedures for each test were administered by the same experimenter. They were also instructed to wear the same footwear during all sessions. To avoid the risk of diurnal variation on performance, all tests were completed at the same time of day (i.e., 4 p.m.–6 p.m.) and under similar environmental conditions (temperature: 19–23 °C, relative humidity: 50%–60%).

#### 2.3.1. Specific Fencing Change of Direction Test

The SFCODT was measured using a 4/2/4/2/4 m as illustrated in Figure 1. The photocells (Brower Timing Systems, Salt Lake City, UT, USA) were placed at the start and at the end of the 8-m distance at hip height (~0.75-m above the ground). The lines were white, 2-m long, and 5-cm wide. For this, fencer athletes started behind one set of timing gates adjusted at hip height (line A). Using specific fencing footwork (the lunge is executed by kicking forward with the front foot and pushing the body forward with the back leg), fencer athletes moved as fast as they could up to the 4-m line (line C), ensured their back foot crossed the line, then moved backward, ensuring the front foot crossed the 2-m line (line B). Again, they moved forward to the 4-m line (line D) before moving backward past line C (2-m long). Finally, they moved forward to the 4-m line (line E). The test was immediately stopped and restarted after 3 min of rest of athletes used fencing steps judged by the evaluators (fencing technician/coach) to be technically incorrect or when athlete failed to cross lines with the two feet while lunging. The test was performed with the participant wearing specific fencing shoes and uniform without holding any weapon. 

#### 2.3.2. Jump Testing 

Vertical jump height was recorded via a portable force platform (Quattro-Jump; Kistler, Winterthur, Switzerland, 500 Hz, V1.1.1.4) through double integration of the vertical reaction force method [19,20,21].

##### Countermovement Jump 

During the CMJ, participants started from an upright erect standing position and performed a fast downward movement by flexing the knees and hips before rapidly extending the legs and performing a maximal vertical jump. During the test, participants were instructed to maintain their arms akimbo. A rest period of 1 min was allowed between trials. The best out of three trials was retained for further analysis. The ICC for test-retest reliability was 0.94.

##### Squat Jump 

The participant started from a stationary semi squatted position (knee angle = 90°) with their hands on the iliac crest jumped upward as high as possible. The intraclass correlation coefficient (ICC)(3,1) for test-retest trials was 0.95.

##### The Five-Jump Test

From a standing position with both feet on the ground, subjects attempted to cover as much distance as possible with five forward jumps by alternating left- and right-leg contact. A tape measure was used to measure the covered distance to the nearest 1-cm. The ICC for test-retest trials was 0.95. 

#### 2.3.3. Sprint and Change of Direction Tests 

*Sprint test.* An electronic timing system (Brower Timing Systems, Salt Lake City, UT, USA) was used to measure the performance of 20-m sprint time. Split sprint times of 5 m, 10 m, and 20 m were analyzed. Players started in a standing start 0.3 m before the first infrared photoelectric gate, which was placed 0.75 m above the ground. The ICCs for test-retest trials were 0.94, 0.96, and 0.97, for 5 m, 10 m, and 20 m, respectively. 

*T-Test.* The protocol outlined by Pauole et al. [16] was used to administer the T-test. Subjects began with both feet behind the starting line. Four cones were arranged in a T-shape, with a cone placed 9.14 m from the starting cone and two more cones placed 4.57 m on either side of the second cone. Each athlete accelerated to a cone and touched the base of the cone with the right hand. Facing forward and without crossing feet, athletes had to shuffle to the left to the next cone and touch its base with their left hand, then shuffle to the right to the next cone and touch its base with their right hand and shuffle back to the left to the last cone and touch its base. Finally, athletes ran backward as quickly as possible to return to the starting/finish line. The performance outcome was collected using an electronic timing system (Brower Timing Systems, Salt Lake City, UT, USA). ICC for test-retest trials was 0.92. 

#### 2.3.4. Isokinetic Strength 

Isokinetic device (Cybex NORM; Henley Healthcare, Cybex International, Inc., Medway, MA, USA) [22] was used to assess isokinetic concentric and eccentric strength of knee extensors and flexors of both limbs. Before testing, each athlete completed a standardized warm-up procedure including 5 min of cycling at 70 W followed by 5 min of hamstring and quadriceps dynamic stretching exercises [23]. Height, limb mass, gravity correction, and individual-specific full range of knee motion were recorded, and each athlete performed a set of 4-5 submaximal leg extensions and flexion contractions at 180 ℃-per-second as a specific warm-up. After 3 min of rest, they were asked to perform five maximal concentric and eccentric repetitions at an angular velocity of 60 ℃-per-second with 3 min of rest between sets. For further analysis, the highest peak torque values (Nm/kg) were used. During continuous (bidirectional) knee extension-flexion movements, the concentric strength of the quadriceps and hamstrings was assessed. A pause of one second was permitted between extension-flexion movements to prevent a contribution to hamstring torque from the previous quadriceps action. During dual concentric-eccentric actions of the same muscle group, eccentric muscle strength was measured. The order of tests was as follows: Concentric extensor, concentric flexor, eccentric extensor, and eccentric flexor. A randomized limb test was chosen for each participant. 

### 2.4. Statistical Analyses

Data were expressed as mean and standard deviation. Measures of normality were assessed using the Shapiro–Wilk statistic, with normality assumed when *p* > 0.05. An independent sample *t*-test was applied to determine significant differences in all variables between sexes and between level groups (high-ranked vs. low-ranked). Effect sizes (ESs) were calculated and judged according to the following scale: ≤0.2, trivial; >0.2–0.6, small; >0.6–1.2, moderate; >1.2–2.0, large; and >2.0, very large [24]. Relative reliability was determined by calculating intraclass correlation coefficient (ICC _(3, 1)_). According to Fleiss [25], ICC can be classified as poor (<0.40), fair (0.40 < ICC < 0.70), good (0.70 < 0.90), and excellent (≥0.90). Absolute reliability was analyzed through the typical error of measurement (TEM) expressed as coefficient of variation (CV). It was calculated by dividing the standard deviation of the difference between scores by √2 [24]. A TEM of less than 5% was set as the criterion for a good absolute reliability (11). To determine any learning effect or systematic bias between SFCODT test and retest scores, a paired samples *t*-test was applied. To establish the sensitivity of the SFCODT, the smallest worthwhile change (SWC_0.2_) was determined [26]. The sensitivity of the test was assessed by comparing the SWC and TEM using the thresholds proposed by Liow and Hopkins [27]. If the TEM was smaller than the SWC, the ability of the test to detect small performance changes was rated “*good*.” If the TEM equaled SWC, then the ability of the test to detect small performance changes was considered “*satisfactory*.” However, in case the TEM was greater than the SWC, the capacity of the test to detect small performance changes was rated *“marginal.”* The minimal detectable change at the 95% confidence interval (MDC_95_) was calculated according to the following formula: MDC_95_ = TEM × 2.77 [28,29]. Discriminant validity was established from the receiving operator characteristic (ROC) curve analysis [30]. According to Deyo and Centor [31], an area under the ROC curve (AUC) >0.70 is commonly considered to indicate acceptable discriminant validity of the test. The association between variables was assessed using Pearson’s correlation coefficients. The magnitude of correlation coefficients was considered as trivial (*r* < 0.1), small (0.1 < *r* < 0.3), moderate (0.3 < *r* < 0.5), large (0.5 < *r* < 0.7), very large (0.7 < *r* < 0.9), nearly perfect (0.9 < *r* < 1), and perfect (*r* = 1) [32]. The level of significance was set *a priori* at *p* ≤ 0.05. All the statistical analyses were computed using SPSS 19.0 program for Windows (SPSS, Inc., Chicago, IL, USA). 

## 3. Results

### 3.1. Comparisons of Women vs. Men and High- vs. Low-Ranked Athletes 

Mean performance scores ± SD at different tests for women and men are detailed in Table 2. The independent sample t-test revealed a significant difference between women and men for vertical and horizontal jumps, 5-m, 10-m, and 20-m sprints, COD, and isokinetic strength peak torque (ES ranged from 1.79 to 3.63 (very large), *p* < 0.001). Figure 2 displays that high-ranked athletes were better than low-ranked on explosive force, COD, and eccentric extensor peak torque (*p* < 0.05, ES = moderate to large). High-ranked fencer athletes showed greater (*p* < 0.01; ES = 1.04) SFCODT performances (7.25 ± 0.41 s) with respect to their low-ranked (7.81 ± 0.67 s) counterparts. Discriminant validity showed an AUC of 0.76 (95% CI: 0.59–0.93; *p* < 0.01). The resulting cutoff for SFCODT performance was <7.78 s (Figure 3).

### 3.2. Absolute and Relative Reliability of SFCODT for Fencer Athletes

The results showed no significant difference between SFCODT test-retest performance (men: *t* = 0.92; *df* = 20; *p* = 0.37; ES = 0.12, Women: *t* = 1.54; *df* = 17; *p* = 0.14; ES = 0.17). The reliability analyses of the SFCODT are shown in Table 3. The results suggest high reliability of SFCODT in both sexes. Specifically, all ICCs values were well above 0.90, while TEM values expressed as CV were <5% (0.06 s and 0.05 s for women and men, respectively). Given that SWC_0.2_ > TEM, the ability of the SFCODT to detect small performance changes can be rated as *“good*.*”* The MDC_95_ values are shown in Table 3.

### 3.3. Correlation between the SFCODT and Components of Physical Fitness

A significant correlation (large to very large) between SFCODT performance and the T-test regardless of sex (men: *r* = 0.75 (R^2^ = 0.56%); women: *r* = 0.79 (R^2^ = 0.62%); combined: *r* = 0.87 (R^2^ = 0.76%)) was observed. In addition, our statistical analyses showed a significant correlation (moderate t -very large) between SFCODT and SJ, CMJ, 5-JT, 5-m, 10-m, 20-m sprints, and isokinetic strength tests (Table 4). The highest correlations obtained were between the SFCODT and mean isokinetic eccentric hamstring force (men: −0.67 to −0.77; women: −0.64 to −0.85).

## 4. Discussion

This study aimed to examine the validity, sensitivity, and the reliability of a new fencing change of direction test among competitive fencer athletes. The main findings of this study were that: i) SFCODT is a reliable (i.e., stable test-retest outcome), useful (i.e., able to detect small changes in performance), and valid test (i.e., significant relationship with *T*-test); and ii) the SFCODT showed significant moderate to very large association with all the selected measures of physical fitness. 

In the present study, the SFCODT showed a high relative reliability associated with a small TEM value irrespective of sex (ICC_(3, 1)_ > 0.90, and TEM < 5%, respectively). In addition, results showed lower TEM values compared with those of SWC_0.2_ in both sexes, indicating the good ability of SFCODT to detect small but meaningful performance changes. Tsolakis, Kostaki, and Vagenas [10] investigated fencer athletes’ COD through a sport-specific test. It consisted of moving as fast as possible with correct fencing steps forward and backward between two lines placed 5-m apart so as to cover a total distance of 30 m. The results from this study reported a high relative reliability with an ICC value of 0.98. However, as fencer athletes’ COD occur over a shorter distance than that proposed by the aforementioned test (i.e., 5 m), the main outcomes may not reflect the real (i.e., “on piste”) COD in fencing. Recently, Turner, Bishop, Cree, Edwards, Chavda, Read, and Kirby [6] proposed another COD test. This test has shown high relative reliability (ICC = 0.95). Although this test seems to better replicate the work time of a fencing compared with that proposed by Tsokalis and Vagenas [5], it can be considered too short with respect to the work time of epee and foil during real competition contests [9]. In this context, the SFCODT took ~7 s to be completed regardless of sex. This duration may be considered closer to the fencer athletes’ means work times [4]. Although the sabre’s work time seems to be slightly less than the half of SFCODT’s duration, the reliability of a COD test that takes less than 3 s would be negatively affected.

The MDC_95_ values established in the present study were 0.12 s for men and 0.14 s for women. These particular findings indicate that a change in SFCODT performance beyond this value could be considered “real” and reflect a true performance improvement in fencer athletes. On the other hand, significant differences were found between SFCODT performance of high-ranked and low-ranked groups. These results were confirmed by the analysis of the AUC derived from the ROC curves. This test is more appropriate to examine the discriminant ability because a difference, although significant, does not necessarily imply that the variable is able to discriminate [33].

Therefore, the significant correlation (55% shared variance) between SFCODT and T-test performances support the criterion-related validity of the SFCODT. Accordingly, the SFCODT can be considered a valid test for assessing COD of fencer athletes. 

The lower-limb power is considered a fundamental prerequisite in fencing [1,3,10], especially for the execution of lunge, as well as for COD performances. Results indicated moderate to large correlations (range: −0.38 to −0.59) between SFCODT and both vertical (i.e., SJ and CMJ) and horizontal jumping (i.e., 5-JT) irrespective of sex. These findings imply that both vertical and horizontal ground-reaction force jumps components may improve COD performance in fencer athletes. 

Sprint time has been considered a major determinant of COD [34,35,36], and the findings of this study support such a contention with moderate to large correlations found between SFCODT and 5-m, 10-m, and 20-m sprint times in both sexes (range: 0.32 to 0.53 and 0.42 to 0.62 for men and women, respectively). With respect to isokinetic strength, moderate to very large correlations were found with SFCODT. The highest correlation obtained was between the SFCODT and mean isokinetic eccentric hamstring strength (*r* = −0.77 and −0.85, respectively for men and women). In accordance with these findings, Anderson et al. [37] reported significant correlations between agility run time and isokinetic concentric and eccentric hamstring force (0.52 to 0.58). It is probable that the eccentric hamstring force was more predictive of agility run time because of the necessity for rapid acceleration and deceleration of the whole body during the activity. In this context, Jones, et al. [38] reported a significant association between COD (i.e., 505 test) performance and hamstring isokinetic eccentric strength in recreational athletes (−0.53 to −0.63). The same authors revealed that sprint time explained 58% of the variance in COD and the addition of eccentric knee flexors strength raised the value to 67%. These results suggest that for basic improvements in COD, athletes should seek to maximize their sprinting ability and to enhance their eccentric knee flexors strength to allow effective neuromuscular control of the contact phase of the COD task. 

Our results support previous research that focused on the relationship between COD and measures of strength with a majority of the evidence supporting the notion that greater muscular strength can enhance change of direction ability [34,39]. Previous studies on the influence of muscular strength on various factors associated with preplanned COD tests were reviewed by Suchomel et al. [39] Though 45 Pearson correlation coefficients initially reported between measures of COD performances and maximal strength (using a variety of multi-joint assessments), 35 of the correlation magnitudes (78%) indicated a moderate or greater relationship with strength, while 27 (60%) displayed a large or greater relationship with strength. This finding is supported by this study, with faster athletes in the SFCODT test producing significantly greater braking and propulsive force compared with slower fencer athletes. Greater force production when changing direction is a combination of superior movement mechanics and strength capacity, resulting in a faster COD performance [40,41]. In addition, increasing force application during the braking phase of COD movements has been shown to increase exit velocity during COD movements [40,42,43] due to an increased storage and utilization of elastic energy as the muscle lengthens under an eccentric load [40,44]. Our results are in accordance to these previous studies reporting moderate to very large associations between lower-body eccentric strength of the rear leg (as measured by isokinetic dynamometry) and SFCODT performance. These results fit into the change-of-direction speed branch of the model presented by Chaabene et al [45], which was modified from Sheppard and Young [46]. According to the model, the eccentric strength of thigh muscles is an important component in COD during the deceleration phase of impulsive movement. Strength and conditioning coaches are recommended to include strength training with accentuated eccentric muscle actions in training routines of fencers to enhance sport-specific performance [45]. Longitudinal studies are needed to determine whether these cross-sectional findings regarding sprint and eccentric muscle strength and COD performance truly reflect deterministic/causative relationships.

This study has some limitations that warrant discussion. First, the ability of fencer athletes to “read and react” (i.e., cognitive components of agility) [46] was not assessed in this study. Therefore, future investigations considering the reaction to a stimulus are needed. Second, future research may consider measuring COD deficit to isolate the COD ability from speed [47]. 

Third, it has to be emphasized that correlations do not establish cause and effect relations. Cross-sectional (i.e., correlational) studies simply show the magnitude of the interrelation between two variables. In other words, a significant relationship between sprint time, concentric strength, and COD performance does not establish cause and effect relations. Therefore, the cross-sectional relationships between variables reported in this study should be interpreted with caution. Future studies should also adopt regression analyses to produce models that combine anthropometric measures with biomechanical variables, as it is likely that both phenotypic and force-related capabilities impact performance [48]. To determine if a relationship is causative, one must investigate changes longitudinally [45,49].

## 5. Conclusions

The SFCODT showed high validity and reliability in fencer athletes. Additionally, the SFCODT showed good ability to detect small but meaningful performance changes. Further, the SFCODT effectively discriminated between fencer athletes from different competitive levels. Moreover, moderate to very large associations were observed between SFCODT and selected measures of physical fitness (i.e., jumping, sprint time, and strength). The latter assumptions, however, should obviously be investigated by future studies. Accordingly, coaches, as well as strength and conditioning professionals, should consider using SFCODT for the purpose of evaluating and monitoring COD performance of their fencer athletes. 

## Figures and Tables

**Figure 1 ijerph-17-04545-f001:**
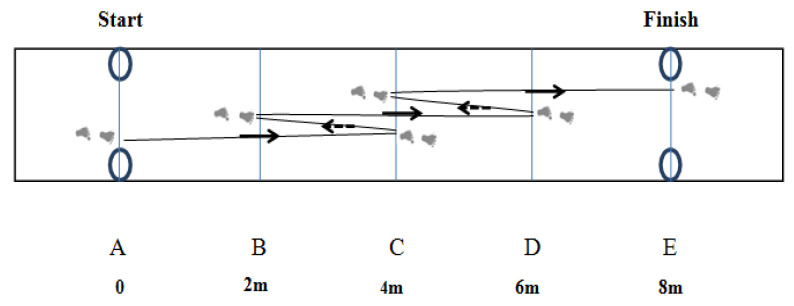
Specific fencing change of direction test (4/2/4/2/4-m).

**Figure 2 ijerph-17-04545-f002:**
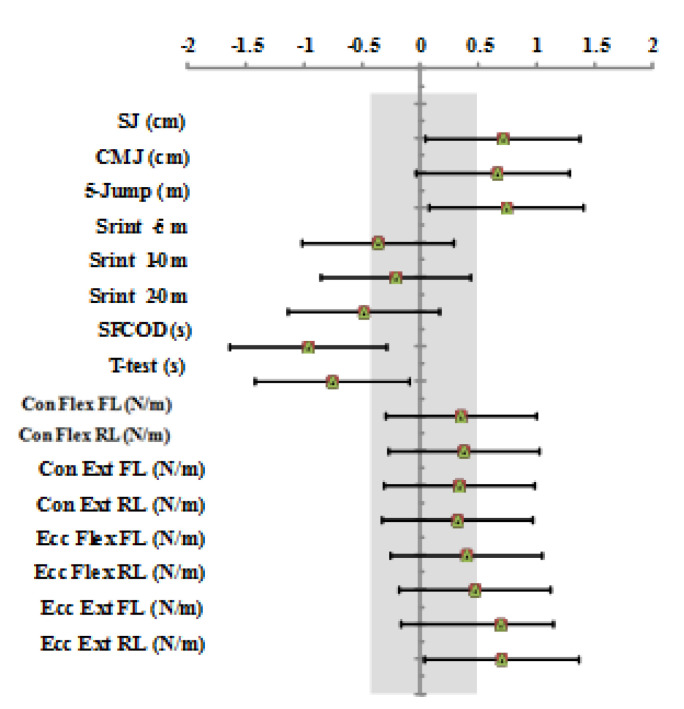
Effect size, with 95% confidence interval, between high- vs. low-ranked fencer athletes.

**Figure 3 ijerph-17-04545-f003:**
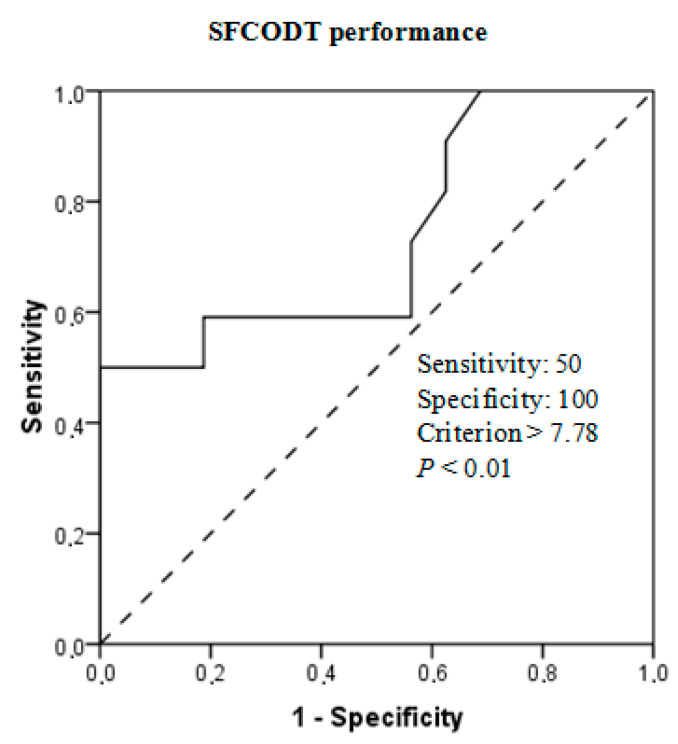
Receiver operating characteristics (ROC) curve for the SFCODT between high- and low-ranked fencer athletes.

**Table 1 ijerph-17-04545-t001:** Anthropometric characteristics of fencer athletes *.

Variables	Women (*n* = 18)	Men (*n* = 21)	Combined (*n* = 39)
Age (year)	19.3 ± 2.5	22.1 ± 2.9 ‡	20.8 ± 3.0
Height (cm)	170.2 ± 3.5	179.9 ± 3.7 ‡	175.5 ± 6.1
Body mass (kg)	63.8 ± 5.9	74.6 ± 7.4 ‡	69.6 ± 8.6
% body fat	22.1 ± 2.3	11.8 ± 3.0 ‡	16.5 ± 5.8

* Values are mean ± SD; ‡ significantly different (*p* < 0.001) for women vs. men.

**Table 2 ijerph-17-04545-t002:** Physical characteristics of fencers *.

Components of Physical Fitness		Women (*n* = 18)	Men (*n* = 21)	Combined (*n* = 39)
**Vertical and Horizontal jump**				
SJ (cm)		39.1 ± 5.1	47.0 ± 4.0 ‡	43.3 ± 6.0
CMJ (cm)		41.5 ± 4.1	50.2 ± 3.5 ‡	46.2 ± 5.8
5-Jump (m)		10.6 ± 0.4	12.6 ± 1.0 ‡	11.6 ± 1.3
**Sprint (s)**				
5-m		1.07 ± 0.02	0.98 ± 0.04 ‡	1.02 ± 0.05
10-m		1.93 ± 0.05	1.69 ± 0.08 ‡	1.80 ± 0.14
20-m		3.34 ± 0.09	3.00 ± 0.13 ‡	3.16 ± 0.21
**CODS (s)**				
SFCODT		7.94 ± 0.36	7.10 ± 0.31 ‡	7.49 ± 0.54
T-test		10.50 ± 0.38	9.80 ± 0.41 ‡	10.12 ± 0.53
**Isokinetic peak torque (N/m)**				
Con Flex	Front Leg	110.7 ± 8.6	157.9 ± 25.3 ‡	136.1 ± 30.7
	Rear Leg	104.7 ± 8.8	150.2 ± 24.0 ‡	129.2 ± 29.5
Con Ext	Front Leg	133.8 ± 17.0	219.6 ± 40.4 ‡	180.0 ± 53.5
	Rear Leg	130.1 ± 15.9	215.9 ± 40.3 ‡	176.3 ± 53.4
Ecc Flex	Front Leg	140.9 ± 13.7	190.0 ± 28.2‡	167.3 ± 33.4
	Rear Leg	130.2 ± 17.7	182.9 ± 29.3‡	158.6 ± 36.1
Ecc Ext	Front Leg	170.5 ± 12.2	251.3 ± 31.2 ‡	214.0 ± 47.4
	Rear Leg	161.5 ± 18.9	244.7 ± 32.9 ‡	206.3 ± 50.0

* Values are mean ± SD; ‡ Significantly different (*p* < 0.001) for women vs. men. Note: CODS = change of direction speed; SFCODT = specific fencing change of direction test; Con Ext = concentric extensor peak torque; Ecc Ext = eccentric extensor peak torque; Con Flex = concentric flexor peak torque; Ecc Flex = eccentric flexor peak torque; SJ = squat jump; CMJ = countermovement jump.

**Table 3 ijerph-17-04545-t003:** Performance characteristics and results of absolute and relative reliability of SFCODT for fencer athletes *.

	Test	Retest	ES	ICC [95% CI]	SEM (s)	SEM (%)	SWC_0.2_ (s)	MDC (s)
Women (*n* = 18)	8.01 ± 0.44	7.94 ± 0.36	0.17	0.93 [0.82–0.98]	0.05	0.60	0.09	0.13
Men (*n* = 21)	7.13 ± 0.27	7.10 ± 0.31	0.12	0.92 [0.81–0.97]	0.03	0.47	0.05	0.09
Combined (*n* = 39)	7.54 ± 0.57	7.49 ± 0.54	0.09	0.97 [0.95–0.99]	0.03	0.38	0.11	0.08

* Values are mean ± SD; ICC = intraclass correlation coefficient; ES = effect size; SEM = standard error of measurement; SWC = smallest worthwhile change; MDC = minimal detectable change.

**Table 4 ijerph-17-04545-t004:** Correlation between the SFCODT and components of physical fitness.

Group	Vertical and Horizontal Jump				Sprint			Isokinetic Peak Torque							
								Con Flex		Con Ext		Ecc Flex		Ecc Ext	
	SJ	CMJ	CMJ-a	5-J	5 m	10 m	20 m	Front Leg	Rear Leg	Front Leg	Rear Leg	Front Leg	Rear Leg	Front Leg	Rear Leg
Women (*n* = 18)	−0.56 (L)	−0.55 (L)	−0.44 (M)	−0.59 (L)	0.62 (L)	0.52 (L)	0.42 (M)	−0.38 (M)	−0.52 (L)	−0.52 (L)	−0.53 (L)	−0.65 (L)	−0.64 (L)	−0.70 (VL)	−0.85 (VL)
Men (*n* = 21)	−0.51 (L)	−0.54 (L)	−0.38 (M)	−0.51 (L)	0.51 (L)	0.53 (L)	0.32 (M)	−0.51 (L)	−0.54 (L)	−0.55 (L)	−0.56 (L)	−0.72 (VL)	−0.67 (L)	−0.77 (VL)	−0.77 (VL)
Combined (*n* = 39)	−0.77 (VL)	−0.82 (VL)	−0.72 (VL)	−0.8 (VL)	0.82 (VL)	0.84 (VL)	0.79 (VL)	−0.77 (VL)	−0.80 (VL)	−0.82 (VL)	−0.82 (VL)	−0.85 (VL)	−0.84 (VL)	−0.89 (VL)	−0.92 (NP)

M = moderate (0.3 < *r* < 0.5), L = large (0.5 < *r* < 0.7), VL = very large (0.7 < *r* < 0.9).

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
