# Peer review of "Validity and Reliability of a New Test of Change of Direction in Fencing Athletes"

_ijerph, 2020, doi:10.3390/ijerph17124545_

Round 1

Reviewer 1 Report

General Comments

The authors need to make clear that their new test also involves a coach to make decisions about the movement patterns in the test. This needs to be stated in the abstract, and noted in the Introduction (i.e. fencer should produce sport-specific movement patterns)

The authors have hyphenated several words that do not need to be hyphenated. Carefully check the manuscript and remove these.

The authors need to explain why isokinetic strength testing was used, and not typical dynamic strength tests used in athletic populations.

Use the term ‘sex’ rather than ‘gender’

There are several instances of font size discrepancies; please check and correct.

A consideration that the authors must note. They state the time for the test is representative of competition because of the length of the test (~7 s). However, and this is also shown through correlations with the T-test and linear sprint times, does this mean that the authors may not be assessing COD ability as closely in their new test? This is because the duration and multiple movement patterns may cover up any deficiencies in COD ability (i.e. a fencer with poor COD ability may be able to make up for it with their explosives or linear speed). The authors must consider this in their discussion, and take note of the work by Nimphius in this regard:

Nimphius, S, Geib, G, Spiteri, T, and Carlisle, D. "Change of direction" deficit measurement in Division I American football players. J Aust Strength Cond 21: 115-117, 2013.

Nimphius, S, Callaghan, SJ, Spiteri, T, and Lockie, RG. Change of direction deficit: A more isolated measure of change of direction performance than total 505 time. J Strength Cond Res 30: 3024-3032, 2016.

Nimphius, S, Callaghan, SJ, Bezodis, NE, and Lockie, RG. Change of direction and agility tests: Challenging our current measures of performance. Strength Cond J 40: 26-38, 2018.

Specific Comments

Abstract

Line 10: ‘change of direction’ should be ‘COD’; you defined the abbreviation

Line 11: ‘jumping ability’ and ‘sprint time’ do not need hyphens. What type of jump and sprint over what distance? This information is needed.

Line 13: how were the rankings determined?

Need to included 12 sentences on the data analysis that was conducted.

Line 20: change ‘can’ to ‘could’

Introduction

Lines 27-28: this is the first time that ‘change of direction’ is used, so define the abbreviation here.

Line 35: change to ‘as a combat sport’

Line 46: change to ‘lines’

Line 54: change ‘tried’ to ‘attempted’

Line 63: insert colon after ‘twofold’

Methods

Line 71: you have already defined the abbreviation ‘COD’ but use the whole term here. Be consistent with the use of the abbreviation, and edit where required throughout the manuscript.

Line 76: insert ‘The’ before ‘T-test’

Line 85: change to ‘participated’

Lines 90-92: provide some specific details as to these accomplishments to reinforce the quality of the participants (i.e. how many medals?)

Line 96: what were the rankings for the players defined as having ‘low rankings”?

The Procedures section should detail how height, body mass, and body fat were measured (and when).

Lines 113-114: again and as stated, ‘sprint time’ and ‘jumping ability’ should not be hyphenated. Secondly, I don’t think the authors should use these catch-all terms for sprinting and jumping. The authors have three different sprint intervals and three different jump tests. They should each be acknowledged separately in case they exhibit different relationships with the SFCODT

Lines 130-131 what was the ‘specific fencing footwork’? Provide specific details.

Line 135: so this test always requires a coach to adjudicate the movement patterns? This is an important consideration, because this now affects the logistics of conducting the test

Line 136: change to ‘The test’

Lines 139-14) this last sentence should be presented earlier in this paragraph

Line 144: ‘squat jump’ does not need to be hyphenated

Lines 151-153: this sentence should be at the start of this section

Line 153: the reference font is larger than the rest of the font here

Lines 155-157: why is the text here italicized?

Line 164: check the reference font size for consistency throughout the manuscript.

Why was isokinetic strength chosen as the strength measure? This needs to be explained and justified. Why not traditional repetition maximum strength testing? This would seem to have greater explanation than the isokinetic strength tests.

Line 204: change to ‘smallest’

Results

Line 223: should refer specifically to the sprint intervals, not just ‘sprint time’

Line 224: check the figure fonts throughout the manuscript; they appear oversized

Line 242: change to “The results’. Same for Line 244.

Line 244: change to ‘suggested’. I would suggest using the term ‘sex’ rather than ‘gender’. If required, make this change throughout the manuscript.

Line 248: change to ‘Table’

Lines 253-254: be specific with ‘jumping ability’ and ‘sprint time’, as you have several different metrics

Discussion

Lines 271-272: check referencing formatting; font size is wrong

Line 274: ‘The results’

Lines 281-283: as noted in the general comments, the length of the test may actually limit how much of the test is influenced by COD ability. The authors should consider this and discuss, as metabolic characteristics and sprint speed could be influencing factors here

Line 288: change to ‘significant differences were found’

Lines 292-294: however, the T-test also incorporates linear sprints, so does not entirely isolate COD ability – which may also be the case in the SFCODT. The authors need to acknowledge and discuss this.

Line 297: correct to ‘correlations’

Lines 301-305: this requires its own paragraph, and discussion re: the points raised by the reviewer (i.e. if sprint time correlates to the SFCODT, does this actually mask COD ability)? This is important to note and discuss.

The paragraph on strength is excessive within the context of the study. Why did other variables not get the attention that strength did? This entire paragraph needs to be cut in half and be more succinct, especially considering there are other major issues that the authors need to address re: their COD test, and how it may (or may not) be measuring COD ability (or just a combination of characteristics important for fencing)

The conclusion should be one paragraph.

Considering the issues raised by this reviewer, after the authors address these they will likely have to edit the conclusion. Same with the Practical Applications section.

Reviewer 2 Report

The article presents an original work. However, I have several methodological, formatting and text overlapping issues.
The methodological issues are focused on the size of the sample. I understand that using a large performance sample from a sport like fencing is complicated, but trying to validate and study the reliability of a test cannot be done with a sample of 39 athletes. I therefore suggest that you carry out a generalisability analysis in order to demonstrate that the sample is adequate. I recommend that you review the following works:
Blanco-Villaseñor, A., Castellano, J., Hernández-Mendo, A., Sánchez-López, C. R. and Usabiaga, O. (2014). Application of the TG in sport for the study of the reliability, validity and estimation of the sample. Journal of Sports Psychology, 23(1), 131-137.
Hernández-Mendo, A., Blanco-Villaseñor, A., Pastrana, J. L., Morales-Sánchez, V., Ramos-Pérez, F. J. (2016). SAGT: Computer application for generalizability analysis. Revista Iberoamericana de Psicología del Ejercicio y el Deporte, 11(1), 77-89.
Regarding the format:
(1) I suggest that you expand the citations of works L.74-75 "In addition, the T-test is frequently used as the most 74 common protocol for testing and validate new tests of planned agility [11-15]".
(2 ) L. 101-103: should add the references of the Helsinki Declaration (WMA 2000, Bošnjak 2001, Tyebkhan 2003).
A report is attached in relation to overlapping text. I believe that the authors should improve the text to avoid this overlap.

Round 2

Reviewer 2 Report

I believe that the authors have made a commendable effort to respond to the issues raised.

However, despite the arguments put forward by the authors, I still consider that they should carry out a generalisability analysis. I believe that the article would be considerably improved.

Despite this issue, the article is of sufficient quality to be published in IJERPH.